# Metformin and Niclosamide Synergistically Suppress Wnt and YAP in APC-Mutated Colorectal Cancer

**DOI:** 10.3390/cancers13143437

**Published:** 2021-07-09

**Authors:** Hee Eun Kang, Yoojeong Seo, Jun Seop Yun, Sang Hyun Song, Dawool Han, Eunae Sandra Cho, Sue Bean Cho, Yoon Jeon, Ho Lee, Hyun Sil Kim, Joyeon Kang, Jong In Yook, Nam Hee Kim, Tae Il Kim

**Affiliations:** 1Department of Oral Pathology, Oral Cancer Research Institute, Yonsei University College of Dentistry, Seoul 03722, Korea; wing870817@gmail.com (H.E.K.); YJS8714@yuhs.ac (J.S.Y.); SSH407@yuhs.ac (S.H.S.); IPODVIDEO@yuhs.ac (D.H.); SANDRA@yuhs.ac (E.S.C.); chosuebean@gmail.com (S.B.C.); khs@yuhs.ac (H.S.K.); jiyook@yuhs.ac (J.I.Y.); 2Department of Internal Medicine, Institute of Gastroenterology, Yonsei University College of Medicine, Seoul 03722, Korea; YJSEO90@yuhs.ac (Y.S.); RKDWH0105@yuhs.ac (J.K.); 3Brain Korea 21 Project for Medical Science, Yonsei University College of Medicine, Seoul 03722, Korea; 4Graduate School of Cancer Science and Policy, National Cancer Center, Goyang 10408, Korea; imyoon81@ncc.re.kr (Y.J.); ho25lee@ncc.re.kr (H.L.)

**Keywords:** colorectal cancer, Wnt, Hippo, niclosamide, metformin

## Abstract

**Simple Summary:**

Hyperactivation of the canonical Wnt and inactivation of the Hippo pathway are well-known genetic backgrounds for familial adenomatosis polyposis (FAP) and colorectal cancer (CRC), although the reciprocal regulation between those pathways is not yet clear. In this study, we found that Axin2, a bona fide downstream target of canonical Wnt, activates the Hippo pathway in APC-mutated CRC, limiting the therapeutic potential of niclosamide on advanced CRC through the inactivation of the Hippo pathway. To overcome the limitation, we combined niclosamide with AMPK activator metformin to activate Hippo and found that this combination synergistically suppressed canonical Wnt and activated Hippo in APC-mutated CRC. Using patient-derived cancer organoid and an APC-MIN mice model, we found the combinatory approach to be effective for APC-mutated CRC. Our results provide not only the reciprocal link between Wnt and Hippo in APC-mutated CRC, but they also provide an effective therapeutic approach with clinically available drugs for FAP and CRC patients.

**Abstract:**

The Wnt and Hippo pathways are tightly coordinated and understanding their reciprocal regulation may provide a novel therapeutic strategy for cancer. Anti-helminthic niclosamide is an effective inhibitor of Wnt and is now in a phase II trial for advanced colorectal cancer (CRC) patients. We found that Axin2, an authentic target gene of canonical Wnt, acts as aYAP phosphorylation activator in APC-mutated CRC. While niclosamide effectively suppresses Wnt, it also inhibits Hippo, limiting its therapeutic potential for CRC. To overcome this limitation, we utilized metformin, a clinically available AMPK activator. This combinatory approach not only suppresses canonical Wnt activity, but also inhibits YAP activity in CRC cancer cells and in patient-derived cancer organoid through the suppression of cancer stemness. Further, combinatory oral administration suppressed in vivo tumorigenesis and the cancer progression of APC-MIN mice models. Our observations provide not only a reciprocal link between Wnt and Hippo, but also clinically available novel therapeutics that are able to target Wnt and YAP in APC-mutated CRC.

## 1. Introduction

Mutations in the APC (adenomatosis polyposis coli) have a well-defined function in canonical Wnt activation and underlie familial adenomatosis polyposis (FAP), an inherited form of CRC that accompanies in 90% of sporadic CRCs [1]. Scaffolding Axin2 is a bona fide downstream target gene of the TCF/LEF transcriptional machinery that is highly expressed in APC-mutant CRCs [2,3,4]. While earlier studies have suggested that Axin is a tumor suppressor, emerging evidence supports the important role of Axin2 in canonical Wnt and the Snail-mediated epithelial–mesenchymal transition (EMT) [5,6,7].

FDA-approved niclosamide has long been used based on its clinical safety and anti-helminthic effectiveness. Interestingly, it was found to have an anti-viral effect against SARS-CoV1 [8,9]. Later, niclosamide was reported to be an effective small molecule inhibitor of canonical Wnt for CRCs both in vitro and in vivo [10]. Subsequently, niclosamide has been reported as a candidate therapeutic in many cancer types, such as breast cancer, ovarian cancer, non-small cell lung cancer, glioblastoma, head and neck cancer, and hematopoietic malignancy [11,12,13,14,15]. Consistent with those anti-cancer effects, several groups have reported that niclosamide suppresses β-catenin mediated canonical Wnt signaling, through means such as inhibited TCF/LEF transcriptional activity and down-regulated pathway target genes, e.g., Axin2 [10,12,16,17]. Recently, Axin2–GSK3 interaction has been proposed as a direct molecular target to support the anti-cancer effects of niclosamide [18]. Currently, phase II trials of re-positioned niclosamide for advanced CRCs (NCT02519582) and COVID-19 based on broad spectrum anti-viral effects are ongoing [19,20].

Evolutionary conserved Hippo signaling inhibits cell proliferation through contact inhibition, the loss of which leads to organ growth and cancer development [21]. The large tumor suppressor (Lats) 1/2 serine/threonine kinase and the AMP-activated protein kinase (AMPK), both of which mediate YAP phosphorylation, are well-known regulators of the Hippo pathway [22], and their inhibition results in increased nuclear YAP activity and TEAD transcriptional activity and increased organ size as well as cancer overgrowth. The YAP–TEAD complex has thus emerged as a novel therapeutic target for human cancer [23,24].

Due to the importance of the Hippo and Wnt pathways in human cancer, a better understanding of the reciprocal link between the two may provide critical insight into human cancer. Recently, we reported that the Wnt scaffolding protein DVL acts as a molecular effector for the nuclear export of phosphorylated YAP in a tumor suppressing context-dependent manner [25], indicating that the Wnt and Hippo pathways are tightly controlled by each other. In this study, we found that the Axin2 scaffolding protein of the Wnt pathway activates the Hippo pathway in APC-mutant CRC cells. Conversely, a knock-down of Axin2 or niclosamide treatment increases nuclear YAP activity, possibly limiting the therapeutic potential of niclosamide for APC-mutated CRC. To overcome YAP activation by niclosamide, we examined metformin as an AMPK activator to re-activate the Hippo pathway. Interestingly, the combination of niclosamide and metformin synergistically suppresses Wnt activity and EMT programs while successfully rescuing nuclear YAP activity in CRC cells and in patient-derived cancer organoids. In vivo, the combination effectively suppresses tumorigenesis and DSS-induced cancer progression in aAPC-Min mice model. Our observations demonstrate that combinatory targeting of the canonical Wnt and YAP pathways may potentiate the therapeutic effect of niclosamide for FAP and APC-mutated CRC patients.

## 2. Materials and Methods

### 2.1. Cell Culture and Transfection

Human colorectal cancer cell lines DLD-1 (ATCC, CCL-221) and human embryonic kidney 293 cells were cultured in Dullbecco’s modified Eagle’s medium (DMEM, Lonza, 12-604F, Basel, Switzerland), and SW480 (Korean Cell Line Bank, KCLB No.10228, Seoul, Korea) was grown in the Roswell Park Memorial Institute 1640 (RPMI 1640, Lonza, 12-702F) with 10% fetal bovine serum (FBS, Life Technologies, Carlsbad, CA, USA) and 100 IU/mL penicillin/streptomycin. SW480 and DLD-1 are colon cancer cells bearing the truncated APC gene. All cells were cultured in a humidified incubator at 37 °C and 5% CO_2_. The Tet-pLKO-puro vector (#21915, Addgene, Watertown, MA, USA) was used for inducible shRNA knockdown. The target shRNA sequence for Axin2 was 5′-ACCACCACTACATCCACCA-3′. Mutant APC expression vectors pCMV-neo-Bam APC 1-1309 (#16508) and pCMV-neo-Bam APC 1-1941 (#16510) were obtained from Addgene. Mycoplasma infection tests were regularly performed with a PCR-based kit (Sigma, MP0040, St. Louis, MI, USA). Transfection was performed using Lipofectamine 2000 according to the manufacturer’s protocols (Invitrogen, 11668-019, Waltham, MA, USA). Niclosamide (2′, 5-dichloro-4′-nitrosalicylanilide) was purchased from CAYMAN, and metformin was obtained from TCI America (TCI, M2009, Portland, OR, USA). The niclosamide and metformin were solubilized in DMSO and distiller water for in vitro experiments.

### 2.2. Immunoblot Analysis

Niclosamide and metformin were treated for 16 h, and the cells were then washed twice with PBS and incubated with 1% triton X-100 lysis buffer (50 mM Tris pH 7.4, 150 mM NaCl, 1 mM EDTA, 1% triton X-100). To evaluate the drug-induced EMT markers, the niclosamide and metformin were treated for 48 h. The cells were collected using a scraper and centrifuged at 13,200 rpm for 15 min at 4 °C. To isolate the nuclear protein fraction, the cells (1 × 10^6^) were collected in microcentrifuge tubes and treated with hypotonic buffer (10 mM HEPES [pH 7.9], 10 mM KCl, 0.1 mM EDTA, 1 mM DTT with protases inhibitors) on ice for 5 min. The cell membrane was ruptured by adding 10% NP-40 to a final concentration of 0.6% and then vigorously vortexed for 10 s followed by high-speed centrifugation for 30 s. The supernatant cytosolic fractions were collected separately, and the nuclear pellets were washed with ice-cold PBS twice. Nuclear protein was extracted with hypertonic buffer (20 mM HEPES [pH 7.9], 0.4 M NaCl, 1 mM EDTA, 1 mM DTT) on ice for 15 min followed by high-speed centrifugation. SDS-polyacrylamide was separated using electrophoresis and transferred to a nitrocellulose membrane (Whatman, Maidstone, UK). After the transfer, the membrane was blocked with 5% skim milk (BD bioscience) for 1 h and incubated at room temperature for 3 h or at 4 °C for 12 h or more. The antibodies against Snail (Cell Signaling, #3895S, Danvers, MA, USA), Axin1 (Cell Signaling, #2087S), Axin2 (Cell Signaling, #2151S), YAP (Santa Cruz, sc-101199), p-YAP(S127) (Cell Signaling, #4911S), S6 (Cell Signaling, #2217S), p-S6(S235/236) (Cell Signaling, #4858S), E-cadherin (BD bioscience, #61081), occludin (Cell Signaling, #5506S), Zeb1 (Cell Signaling, #D80D3), fibronectin (Abcam, ab2413, Cambridge, United Kingdom), tubulin (AB Frontier, LF-PA0146A, Seoul, Korea), and HDAC1 (Santa Cruz, sc-81598) were obtained from commercial vendors. Phos-tag gel was purchased from WAKO chemicals (AAL-107).

### 2.3. Cell Migration Assay

For migration assays with niclosamide and metformin, DLD-1 and SW480 cells (5 × 10^4^) were seeded into transwell inserts (5.0 μm pore, BD Biosciences). The filter inserts were prewetted before the cells were added with PBS. 1 mL of medium was added to the lower chamber. Cells were added to 5 × 10^4^/100 μL medium in the top of the inserts. After a 48 h culture period in the medium, which contained 0.5 μM niclosamide or 10 mM metformin or a combination of the two was added to the bottom, the cells were washed twice with PBS and fixed with 4% formaldehyde. The upper part was wiped with cotton, and the cells in the lower part were stained with 0.25% crystal violet. Cell counts were determined in five random fields.

### 2.4. Quantitative Real Time-PCR and Reporter Assay

Total RNA was extracted with TRIzol reagent (Invitrogen) according to the manufacturer’s protocols. cDNA was generated using the SuperScript III Synthesis Kit (Invitrogen). Real-time quantitative PCR (qPCR) analysis was performed with the ABI-7300 according to the SYBR Green Mix protocol (n = 3). Each ΔCt value from each sample was calculated by normalization with GAPDH. Primer specificity was confirmed by the dissociation curve after qPCR reaction. Primer sequences used for real-time qPCR are shown in Appendix A. For TEAD or TCF/LEF transcriptional activity, the cells were transfected with 100 ng of the reporter vectors and 1 ng of the pSV-Renilla expression vector. Luciferase and renilla activity were measured using the Dual-Luciferase Reporter System Kit (Promega, Madison, WI, USA) 48 h after transfection and normalized through renilla activity. Results were averaged by triplicate experiments.

### 2.5. Immunofluorescence

For the immunofluorescence studies, cells were washed twice with ice cold PBS and incubated with 4% formaldehyde for 15 min at room temperature. For staining, the cells were permeabilized with 0.5% Triton X-100 for 45 min, blocked with PBS containing 3% bovine serum albumin for 1 h, and then incubated with primary antibody overnight at 4 °C. Cells were then washed three times with PBS containing 0.1% Tween 20 followed by incubation with anti-mouse-Alexa Fluor-594 (Invitrogen, A11005) secondary antibody. Cells were mounted with the Vectorshield (Vector Laboratories, Burlingame, CA, USA) with 4′,6-diamidino-2-phenylindole (DAPI). Cellular fluorescence was monitored using confocal microscopy (Zeiss LSM780, Jena, Germany). Images were analyzed with the assistance of Image J software (Media Cybernetics, Inc., Rockville, MD, USA). The relative fluorescence intensity was measured in the whole nuclear area.

### 2.6. Gene Expression Analysis of Clinical Samples

Publicly available mRNASeq data of 220 colorectal cancer samples (COADREAD) having APC or CTNNB1 mutational status from The Cancer Genome Atlas (TCGA) were downloaded (https://gdac.broadinstitute.org (accessed on 2 February 2017)). The illuminahiseq_rnaseqv2-RSEM_genes_normalized (MD5) was log2 transformed, and the relative transcript abundance of Axin2, CTGF, and CYR61 were compared using Pearson correlation. Separately, a reverse phase protein arrays (RPPA) dataset of 483 COADREAD from TCGA was obtained. The YAP and active AMPK protein abundance were compared using Pearson correlation. The scatter plots of CTGF and Axin2 abundances were obtained using R.

### 2.7. Culture of Patient-Derived Organoids from Colon Cancer and FAP Patients

Patient-derived organoid culture experiments were approved by the Institutional Review Board of Yonsei University College of Medicine, Severance Hospital (IRB No. 4-2012-0859), and endoscopic biopsy samples were obtained from patients with informed consent. The biopsy samples were washed in PBS with 100 μg/mL primocin (InvivoGen, San Diego, USA) and chopped into pieces of around 0.5 mm and further incubated in digestion buffer (DMEM, 2.5% FBS, 6.25 mg/mL collagenase type IX (Sigma, St. Louis, MO, USA)) for 30 min at 37 °C. Next, the isolated cells were mixed with 20 μL of Matrigel (growth-factor reduced, phenol red free; BD Bioscience) and plated in 48-well plates. After the polymerization of the Matrigel, tumor organoid culture medium (advanced DMEM/F12 with 1% penicillin/streptomycin (Glutamax); 1 × N2 and 1 × B27 without retinoic acid(all from Invitrogen); 2 mM L-glutamine (Life Technologies); 50 ng/mL EGF (R&D Systems); and 1 mM N-Acetyl-L-cysteine, 10 mM Nicotinamide, 10 nM Gastrin I, 2.5 μM PGE2 and 100 ng/mL Noggin (all from Sigma)) was overlaid (250 μL/well). For the first 2 days after plating, the medium was also supplemented with 10 μM ROCK inhibitor Y27632 (Sigma), and the culture medium was changed every 2 days. For passage, the organoids and Matrigel were washed in PBS and dissociated with trypsin/EDTA (Sigma). The dissociated organoids were embedded in 20 μL of Matrigel and seeded in 96-well plates. The Matrigel was polymerized and then overlaid with 100 μL/well basal culture medium with niclosamide (0.5 or 1 μM), metformin (2 mM), or a combination of niclosamide and metformin. The culture medium was changed every 2 days.

### 2.8. Tumor Sphere Culture

SW480 and DLD-1 cells (2000 cells per well) were plated in 24-well ultra-low adhesive plates (Corning Incorporated, Corning, USA) in sphere formation media with niclosamide (1 μM), metformin (2 mM), or a combination of niclosamide and metformin for 8 days. The sphere formation media were serum-free DMEM-F12 supplemented with B27 (Life Technologies, Carlsbad, CA, USA), 20 ng/mL epidermal growth factor (EGF), 10 ng/mL basic fibroblast growth factor (R&D Systems, Minneapolis, MN, USA), 1% penicillin/streptomycin, and 2 mM L-glutamine (Life Technologies, Carlsbad, CA, USA). At the end of the experiment, the cells were stained with calcein AM and, the live cell fluorescence intensities of tumor spheres were then measured using ImageJ.

### 2.9. Flow-Cytometric Analysis for Cancer Stem Cell Markers

Colorectal cancer cell lines SW480 and DLD-1 were plated at a density of 2 × 10^5^ cells/well in 6-well plates and treated with niclosamide (0.1 μM), metformin (10 mM), and a combination of niclosamide and metformin. After 48 h, the prepared cells were detached by Accutase (Millipore, Billerica, MA, USA) and resuspended in FACS buffer (1 × PBS, 1% bovine serum albumin, and 2 mM ethylene diamine tetra-acetic acid). Antibodies for cancer stem cell markers (PE-conjugated anti-CD166 and FITC-conjugated anti-CD44) were added and incubated for 10 min at 4 °C. The samples were then washed with FACS buffer and subjected to flow cytometry for analysis using a FACSVerse (BD Biosciences, San Diego, CA, USA) coupled to a computer with BD FACSuite software.

### 2.10. APC-MIN Mouse Model for FAP and DSS-Induced Colon Cancer

APC min mice were produced by mating C57BL/6J wild type female mice with C57BL/6J-APC min +/− (APC min, The Jackson Laboratory strain 002020) male mice. When the APC min mice reached 6 weeks of age, chemical treatment was started. For the combination treatment of niclosamide and metformin, the mice were treated with vehicle or with niclosamide (50 mg/kg, P.O.) only or with metformin (2 mg/mL, P.O.) only or both niclosamide (50 mg/kg, P.O.) and metformin (2 mg/mL, P.O.) for 14 weeks, after which the mice were sacrificed, and the entirety of their intestines were dissected. The tissues were fixed in 4% formaldehyde for 24 h and then washed twice with 70% ethanol. The number of polyps in the small intestine was calculated using a stereomicroscope. Polyps under 1 mm were small, those that were between 1 mm and 3 mm were medium, and polyps that were over 3 mm were large.

For the mouse colon cancer model, six-week-old APC min/+ male mice were treated with drinking water containing 3% DSS (*w*/*v*) for 6 days, followed by a 4-week recovery period with regular water. The DSS-treated APC min/+ mice were then randomly divided into four groups (5 mice per group): control, metformin (metformin, 500 mg/kg/day in drinking water), niclosamide (niclosamide, 50 mg/kg/day in diet), and combination (metformin and niclosamide). The mice were monitored daily and weighed thrice weekly, with the amount of drinking water and their diets being measured daily. Mice were sacrificed after 3 weeks of treatment with metformin or niclosamide or a combination of the two, and intestinal tissue samples were collected for further analysis. The opened intestinal segments were spread flat between sheets of filter paper and stained with 0.2% methylene blue. The stained sections were rinsed in deionized water and imaged by camera, and the tissue was then fixed with 4% PFA for 24 h. These experiments were performed in accordance with protocols approved by the Institutional Animal Care and Use Committee of the Yonsei University.

### 2.11. Immunohistochemistry for In Vivo Samples

IHC was performed on 4 μm sections of formalin-fixed, paraffin-embedded, dissected tumor samples for CD166, CD44, pS6, and β-catenin. The paraffin-embedded sections were deparaffinized in xylene and rehydrated in gradually decreasing concentrations of ethanol. Antigen retrieval was performed using sodium citrate buffer (10 mM, pH 6.0) in a heated pressure cooker for 5 or 7 min. After incubation with 3% hydrogen peroxide to block endogenous peroxidase activity for 30 min, the sections were incubated in a blocking reagent for 30 min at room temperature. Anti-CD166 antibody (1:200 dilution), anti-CD44 antibody (1:100 dilution), anti-pS6 antibody (1:100 dilution), and β-catenin (1:6000 dilution) were incubated with the sections overnight at 4 °C, and the secondary antibody was then incubated for 30 min at room temperature. After the slides were developed with a Vectastain ABC Kit (Vector Laboratories), immunostaining was performed using DAB solution (Dako, Carpinteria, CA, USA). After counterstaining with hematoxylin, IHC staining was evaluated by light microscopy, and immunoactivity was assessed based on the proportion of immunostained tumor cells. We measured the intensity of CD44, CD166, pS6, and β-catenin through IHC scores using IHC profiles based on the ImageJ program [26]. The calculation for IHC score is ‘Score = (number of pixels in a zone) × (score of the zone)/total number of pixels in the image’. The assigned scores were 4 for the high positive zone, 3 for the positive zone, 2 for the low positive zone, and 1 for the negative zone.

### 2.12. In Vivo Xenograft Assay

All animal experiments were conducted by the Institutional Animal Care and Use Committee of Yonsei University and approved by the Animal Care Committee of Yonsei University School of Dental Sciences and the National Cancer Center Research Institute. Female BALB/c nude mice (6 weeks old, purchased from Nara Biotech, Seoul, Korea) were used for xenograft assays with subcutaneous injection. To study the anti-tumor effect of combined niclosamide and metformin, SW480 cells in the control or experimental groups were trypsinized, harvested, and injected into subcutaneous tissue (1 × 10^6^ cells per 0.1 mL PBS). For the combination treatment of niclosamide and metformin in vivo, the mice were treated either with vehicle, with niclosamide (200 mg/kg, P.O.) only, with metformin (2 mg/mL, P.O.) only, or with both niclosamide (200 mg/kg, P.O.) and metformin (2 mg/mL, P.O.) after the subcutaneous injection of SW480 cells. These treatments were applied 5 times a week for 4 weeks from the day after the mice received the tumor cell injections. Tumor growth and body weight were monitored twice a week with Vernier calipers, and tumor volume was calculated with the equation V (in mm^3^) = (a × b^2^)/2: a, the longest diameter; b, the shortest diameter. The proliferation index was examined using Ki-67 immunostain, and relative Ki-67 positive abundance was determined by the ImageJ program downloaded from NIH (https://imagej.nih.gov/ij/ (accessed on 23 August 2019)).

### 2.13. Statistical Analysis

All statistical analyses of cell viability, cell migration, and reporter assay were performed with two-tailed Student’s *t*-tests; data are expressed as means and s.d. Triple asterisks denote *p* < 0.001, double asterisks denote *p* < 0.01, and one asterisk denotes *p* < 0.05. Statistical significance of animal experiments was determined using the Mann–Whitney test; data are expressed as mean and s.e.m. for tumor volume. No statistical method was used to predetermine the sample size.

## 3. Results

### 3.1. Axin2 Potentiates the Hippo Pathway in APC-Mutant Colorectal Cells

The regulatory axis of canonical Wnt-dependent Axin2 plays an important role in the regulation of Snail (SNAI1) mediated EMT in cancer [5], is a typical downstream target of the TCF/LEF transcript factor, and is highly abundant in CRC due to the loss of APC [3,4]. To determine the role of Axin2 on the Hippo pathway in clinical samples, we analyzed RNA expression data from primary human CRC having APC/CTNNB1 mutational status (210 samples from The Cancer Genome Atlas, TCGA), choosing CTGF, CYR61, and Axin2 transcripts, which are representative transcriptional target genes of the YAP and TCF/LEF machinery, respectively [27]. Unlike in breast cancer samples [25], Wnt activity determined by Axin2 was inversely correlated to YAP-mediated transcriptional activity in CRC samples, particularly in APC or CTNNB1 mutated cancer samples (Figure 1A). In the CRC cancer patients, abundances of YAP-mediated transcripts were associated with a worse prognosis while the association with Axin2 was inverted (Appendix A). Because Axin2 is abundant in CRC cells, we next examined the protein abundance of Axin and the YAP phosphorylation status in APC-mutated CRC cell lines. Consistent with previous observations [3,4], the Axin2 protein is highly expressed in APC or CTNNB1-mutated CRC cell lines while Axin1 was equivalently expressed regardless of APC or CTNNB1 mutational status (Appendix A). Notably, YAP phosphorylation status was correlated with Axin2 abundance, suggesting that Axin2 may activate the Hippo pathway in APC-mutated CRC. To directly determine the role of Axin2 in the Hippo pathway, we chose two CRC cell lines with APC mutation and made inducible Axin2 knockdown cells. Interestingly, the knockdown of Axin2 in CRC cells suppressed YAP phosphorylation as determined by pSer127-YAP and mobility shift while the total YAP level increased (Figure 1B). Along with decreased YAP phosphorylation, the CTGF transcript level and TEAD reporter activity were increased by knockdown of Axin2. Consistent with previous observations [5,6], the knockdown of Axin2 suppressed EMT-inducer Snail along with increased E-cadherin abundance. Considering that E-cadherin/α-catenin tumor suppressors strongly induce YAP phosphorylation as an upstream regulator of the Hippo [25,28], these results indicate that Axin2 specifically inhibits the Hippo pathway in CRC. To directly address this hypothesis, we transfected mutant APC into 293 cells together with Axin1 or Axin2 and examined the YAP phosphorylation status. Indeed, Axin2 increased YAP phosphorylation, especially the in APC-mutant background (Figure 1C). Non-sense mutations in C-terminal of Axin2 are frequently found in CRC with defective mismatch repair [29]. When we examined the effect of wild type (wt) and C-terminal truncated Axin2 on YAP activity in the 293 tested cells, wt Axin2 consistently suppressed TEAD reporter activity while C-terminal truncated Axin2 largely relieved reporter activity (Appendix A). These results indicate that Axin2 inhibits the tumor-suppressive Hippo pathway, resulting in increased nuclear YAP activity in APC-mutated CRCs.

### 3.2. Niclosamide Suppresses the Canonical Wnt and Hippo Pathways in APC-Mutant Cells

While niclosamide has emerged as a CRC therapeutic due to its suppression of the Wnt pathway and Axin2/Snail-mediated EMT program [10,16,18], its effect on Hippo has not yet been determined. We therefore treated the CRC cells with niclosamide and examined YAP phosphorylation status and transcriptional activities. The niclosamide decreased Axin2 and Snail abundance along with TCF/LEF transcriptional activities (Appendix A). We further evaluated the protein levels YAP and p-YAP in 293 cells, which were not APC-mutant and not-tumor cells. YAP phosphorylation was not affected by niclosamide in those 293 cells (Appendix A). Given that Axin2 increases YAP phosphorylation in CRC, the niclosamide increased nuclear YAP abundance by inhibiting its phosphorylation in APC-mutated CRC cells (Figure 2A). Niclosamide also increased endogenous YAP transcriptional activity by means of CTGF transcript abundance and TEAD reporter assay (Figure 2B). In an immunofluorescence study, niclosamide significantly increased the nuclear YAP abundance in CRC cells (Figure 2C). The p53 and Lats2 tumor suppressors comprise a feed-forward loop, and the loss of wild type p53 governs DVL-mediated nuclear YAP activity [25,30]. To determine the effect of the p53 tumor suppressor on niclosamide, we examined the YAP phosphorylation status in wt and p53-null HCT116 cells and in CRC cells. Interestingly, niclosamide decreased YAP phosphorylation regardless of the wt p53 status in the HCT116 cells as well as in the p53-mutated SW480 cells (Figure 2D). These results indicate that niclosamide inhibits tumor suppressive Hippo followed by oncogenic YAP activation although it effectively suppresses canonical Wnt activity, suggesting the potentially limited clinical effectiveness of niclosamide in CRC patients (Figure 2E).

### 3.3. AMPK-Activator Metformin Attenuates Niclosamide-Mediated Hippo Suppression

In the Hippo pathway, the large tumor suppressor (Lats) kinase phosphorylates at multiple YAP sites, including Ser127, regulating nuclear YAP activity [31]. Independently, metabolic AMP-activated protein kinase (AMPK) has been found to directly phosphorylate YAP, resulting in decreased oncogenic YAP activity in a DVL-dependent manner [22,25,32]. Metformin is widely used and is a well-known activator of AMPK [33]. Thus, we chose metformin to overcome niclosamide-mediated Hippo inactivation in APC-mutated CRC. Consistent with previous observations [22,25,32], metformin not only effectively suppressed nuclear YAP activity by means of increased YAP phosphorylation in APC-mutated CRC cells, but also moderately suppressed canonical Wnt activity, the Snail-mediated EMT program, and mTOR activity (Appendix A), indicating that metformin can attenuate niclosamide-mediated oncogenic YAP activation as well as suppress Wnt activity.

Prior to testing the metformin on YAP activity, we examined the combination of niclosamide and metformin on the canonical Wnt and Snail-mediated EMT programs. Interestingly, metformin potentiates the effect of niclosamide on these programs by means of Axin2 and Snail abundance in APC-mutated CRC cells as well as in their transcriptional program (Figure 3A,B). When we examined the transcript abundances and protein levels of epithelial and mesenchymal markers, the combination of niclosamide and metformin significantly increased epithelial genes while suppressing mesenchymal markers (Figure 3C,D). Accordingly, the combination effectively inhibits the transwell migration of CRC cells (Figure 3E), indicating that metformin potentiates the niclosamide-mediated suppression of Wnt activity and the Snail-mediated EMT program.

Because oncogenic YAP stability and activity are directly regulated by AMPK, we next explored the effect of AMPK activity on YAP abundance in clinical CRC samples. In the TCGA dataset, protein expression levels as determined by reverse phase protein arrays (RPPA) were available for 483 CRC patients (COADREAD). By analyzing active AMPK and YAP abundance, we found that active AMPK was inversely correlated to YAP abundance in the CRC patient samples (Figure 4A), indicating that AMPK activity plays an important role in oncogenic YAP stability and suggesting the potential role of the AMPK activator in Hippo in CRC. Given the Hippo activation by metformin, we next examined whether metformin positively influences niclosamide-mediated Hippo suppression. Indeed, metformin increased YAP phosphorylation, resulting in decreased nuclear YAP abundance (Figure 4B), as well as suppressing the CTGF transcript abundance increased by niclosamide (Figure 4C). Interestingly, niclosamide or metformin alone can activate AMPK, and the combination of the two significantly increased phosphorylated AMPK under physiologic glucose conditions. Consistent with the opposing actions of AMPK and mTOR (target of rapamycin) in cell growth control [34], metformin also suppressed mTOR activity in combination with niclosamide in CRC cells (Appendix A). An immunofluorescence study further supports the role of metformin in intracellular YAP translocation (Figure 4D). These results show that the AMPK activator metformin successfully positively influences Hippo suppression caused by the Wnt inhibitor niclosamide in APC-mutated CRC.

Given the observation that niclosamide suppresses the Hippo pathway, we next tested whether metformin could attenuate oncogenic YAP activity induced by niclosamide treatment. To test the effect of the combination of the two on tumorigenesis, we next used an in vivo xenograft model. The athymic nude mice who had been subcutaneously injected with SW480 cells were orally administrated either niclosamide (200 mpk), metformin (2 mg/mL in drinking water), or a combination of the two, and the tumor volume was monitored for 40 days. Interestingly, the tumor formation and volume significantly decreased with the single or combinational administration of niclosamide and metformin (Appendix A), while the body weight of the mice was unchanged. To further examine the xenografted tumor samples, we assessed the proliferative potential of an in vivo CRC tumor by means of a immunohistochemical stain for Ki-67. Interestingly, niclosamide or metformin decreased the in vivo proliferation of the CRC cells, and the combination synergistically inhibited the Ki-67 index in vivo (Appendix A). To further examine the clinical relevance of the combinatory approach in the APC-mutated context, we used FAP-patient derived polyp organoids. Niclosamide and metformin separately induced the suppression of the FAP organoids while their combined treatment yielded the further decrease of the FAP organoids (Figure 4E). These results indicate that the AMPK activator metformin potentiates the anti-tumor effect of niclosamide for FAP patients through the suppression of canonical Wnt and mTOR activity as well as by the activation of the tumor suppressive Hippo pathway (Figure 4F).

### 3.4. The Combination of Niclosamide and Metformin Suppresses Tumor Organoids and the In Vivo Tumor Potential of CRC

The signaling pathways of Wnt, Yap, and mTOR play important roles in maintaining cancer stem cells in colorectal cancer, and it has recently been discovered that node signaling may also play a role [35]. Therefore, we tested the effect of niclosamide, metformin, and a combination of the two on the tumor sphere and cancer stem cell population in CRC cells and patient-derived tumor organoids. In SW480 and DLD-1 cells, the combination of niclosamide and metformin yielded a significant decrease of tumor spheres and a lower proportion of CD44+/CD166+ cells compared to those of cells treated with niclosamide or metformin alone (Figure 5A,C and Appendix A). In addition, when using established patient-derived cancer organoids, we found the same effect of those drugs, as shown in Figure 5B, indicating that Wnt inhibition and Hippo activation caused by the combination treatment effectively suppresses the stemness of CRC cells and the patient-derived samples.

To determine the therapeutic potential of niclosamide and metformin, we next used an APC-MIN (APC^Δ850^) mice model. APC-MIN mice (3 weeks old) were orally administered both niclosamide and metformin in daily doses of 50 mg/kg niclosamide and 2 mg/mL metformin in drinking water. Fourteen weeks after administration, the intestinal adenoma burden was significantly decreased, with body weight being unaffected (Figure 5D).

### 3.5. Combination of Niclosamide and Metformin Efficiently Suppresses Cancer Progression in APC-MIN-DSS Model

Finally, we investigated the effect of the combinatory targeting of Wnt and YAP in a well-established DSS-induced colon cancer model using APC-MIN mice (Figure 6A). While the oral administration of metformin or niclosamide alone was minimally effective compared to the control, their combination significantly suppressed colon tumor progression in the APC-mutant model in terms of the number and area of tumors (Figure 6B–D). Immunostaining for β-catenin and phosphor-S6 showed that niclosamide suppressed β-catenin expression while the metformin suppressed mTOR activity in cancer tissue (Figure 6E,F). Further evaluating the colon cancer stem cell (CSC) markers, we found that both CD44 and CD166 expression were reduced in the DSS-induced colon cancer samples of mice treated with the same drugs, particularly the combination of niclosamide and metformin (Figure 6G,H). These results indicate that the combinational suppression of canonical Wnt and the activation of AMPK-dependent Hippo with clinically available drugs effectively suppresses CRC progression in an APC-mutated genetic background.

## 4. Discussion

Hyperactivation of oncogenic Wnt caused by APC mutation is well-known as genetic background for human FAP and CRC. As a typical transcription target gene of canonical Wnt, Axin2 is highly expressed in CRC [3,4]. Although Axin1 has been widely regarded as a tumor suppressor, Axin2 plays a key regulatory function in the Snail-mediated EMT program and cancer progression caused by the nuclear-cytosolic shuttling of GSK-3 [5,6]. Together with canonical Wnt, inactivation of the tumor-suppressive Hippo pathway resulting in YAP activation comprises fundamental signaling involved in many types of human cancer [21]. Although the Wnt and Hippo pathways share intercellular adhesion and nuclear transcriptional programs, the reciprocal link between those pathways is not well understood. According to the open public resources in Gepia, the expression level of Axin2 is markedly abundant in CRC tissues compared to normal colon tissues, and the expression ratio in CRC samples in respect to normal tissues of Axin2 is inversely correlated with the ratio of YAP-mediated transcripts. A recent study proposed that the Wnt scaffolding protein DVL is responsible for the intracellular trafficking of YAP phosphorylated caused by Lats or AMPK [25]. In this study, we found that Axin2 activates the Hippo pathway, especially given the APC-mutated genetic background of CRC. Conversely, targeting Axin2 with shRNA or niclosamide suppresses the Hippo pathway while effectively suppressing canonical Wnt in CRC.

Niclosamide is an orally administered drug that was approved by the FDA in 1982 [36] and designated as a World Health Organization (WHO) essential drug in developing countries. Because it also has broad spectrum antiviral effects against SARS-CoV, MERS-CoV, Zika virus, and human adenovirus [37], clinical trials for its use against SARS-CoV2 are ongoing in multiple centers (NCT04542434, NCT04399356, NCT04558021). Recent studies have shown that niclosamide can treat cancer by controlling several signaling pathways such as the Wnt, STAT3, and Notch pathways [16,18,38,39]. In particular, niclosamide is known to be a potent canonical Wnt pathway inhibitor in human ovarian cancer, breast cancer, retinoblastoma, prostate cancer, and colorectal cancer [12,18,40,41,42]. Based on these observations, a phase II trial of niclosamide in the treatment of advanced CRCs is ongoing [19]. Niclosamide exerts a pharmacological effect by disrupting the Axin–GSK3 protein–protein interaction, at least in part, resulting in the suppression of Axin2 and the Snail-mediated EMT program [18]. In this study, we found that niclosamide activates nuclear YAP activity with inactivation of the tumor-suppressive Hippo pathway in APC-mutated CRC cells. These results suggest a potential therapeutic limit of niclosamide in the clinical trial of CRC patients because YAP activation is an independent predictor of prognosis in CRC [43].

Overexpression of oncogenic YAP has been associated with several types of cancer including lung, colorectal, and breast cancer [44,45,46]. In particular, YAP is required for CRC development in APC-mutant CRC [47], and the inactivation of the Hippo pathway accompanies high metastatic potential and poor survival outcome [43]. The AMPK is a key kinase in the Hippo pathway because of the multiple phosphorylations of YAP, including Ser127 [22,32]. Interestingly, AMPK plays an important role in regulating YAP stability in clinical CRC samples in the TCGA dataset. Since pharmacological activation of Lats kinase is currently very limited, we chose metformin as a clinically available AMPK activator to overcome the therapeutic limitation of niclosamide in APC-mutated CRC [33].

The biguanide compound metformin is the most common therapeutic for type 2 diabetes. Several large-scale cohort studies have revealed that the long-term use of metformin decreases the risk of CRC and colorectal adenoma formation [48,49]. Furthermore, high-dose metformin effectively suppressed intestinal polyp formation in a APC-MIN mice model [50], revealing that metformin is effective for the chemoprevention of CRC. Mechanistically, metformin is a potent AMPK activator and suppresses the mTOR pathway as well as EMT in CRC cells [51]. In this study, we showed that metformin in combination with niclosamide effectively suppresses the canonical Wnt and Snail-mediated EMT programs as well as activates the Hippo pathway in APC-mutated CRC. Our study provides a novel therapeutic option for CRC and FAP patients by targeting both oncogenic Wnt and YAP.

## 5. Conclusions

The activation of Wnt and the inactivation of Hippo comprise key signaling pathways in cancer. Kang et al. demonstrate that the combination of the clinically available drugs niclosamide and metformin synergistically inhibits APC-mutant CRC progression via the suppression of Wnt and YAP.

## Figures and Tables

**Figure 1 cancers-13-03437-f001:**
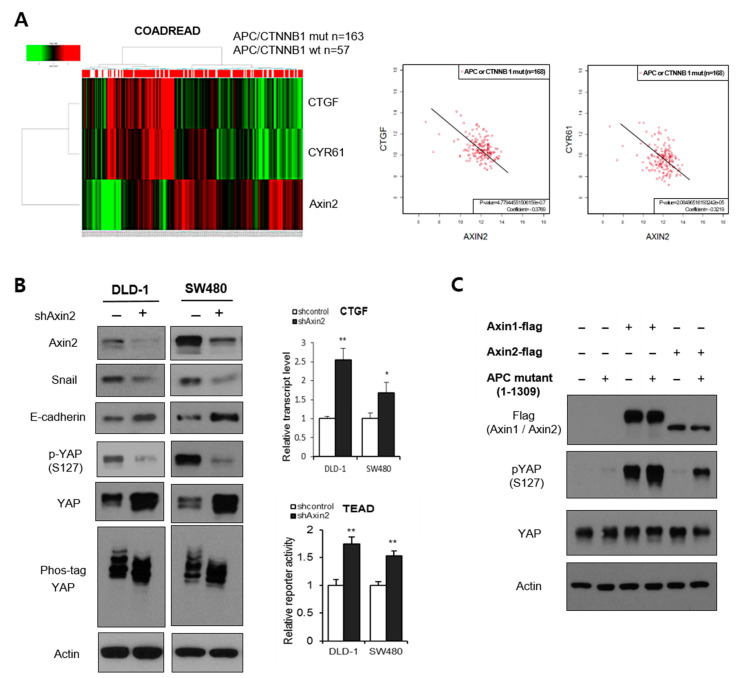
Axin2 potentiates the Hippo pathway in APC-mutant colorectal cancer. (**A**) Axin2 transcript abundance in CRC patient samples was inversely correlated to YAP target genes CTGF and CYR61. Those transcript abundances in human COADREAD samples from the TCGA Illumina HiSeq database are represented in a heatmap (left panel, Axin2 vs. CTGF, *p* = 2.456 × 10^−7^; Axin2 vs. CYR61, *p* = 4.068 × 10^−6^, Pearson correlation). APC or CTNNB1 mutant samples are denoted as red bars. Scatter plots of CTGF, CYR61, and Axin2 transcript in TCGA COADREAD samples having APC or CTNNB1 mutation (n = 168). Axin2 vs. CTGF, R2 = −0.3769, *p* = 4.779 × 10^−7^; Axin2 vs. CYR61, R2 = −0.3219, *p* = 2.0848 × 10^−5^. (**B**) Inducible knockdown of Axin2 suppressed Hippo pathway by decreased YAP phosphorylation. YAP phosphorylation level was determined by pS127-YAP antibody and mobility shift on a phos-tag gel in inducible knockdown of Axin2 with doxycycline (Dox) treatment 3 μg/mL for 48 h (left panels). Relative CTGF transcript abundance and TEAD reporter activity of CRC cells expressing inducible shRNA of Axin2 (right). Statistical significances compared to control are denoted as * *p* < 0.05; ** *p* < 0.01 by a two-tailed Student’s *t*-test. (**C**) Axin2 increased YAP phosphorylation in the presence of APC mutation. 293 cells were transfected with indicated vectors, and total YAP and phosphorylation YAP (p127-YAP) were determined by Western blot analysis.

**Figure 2 cancers-13-03437-f002:**
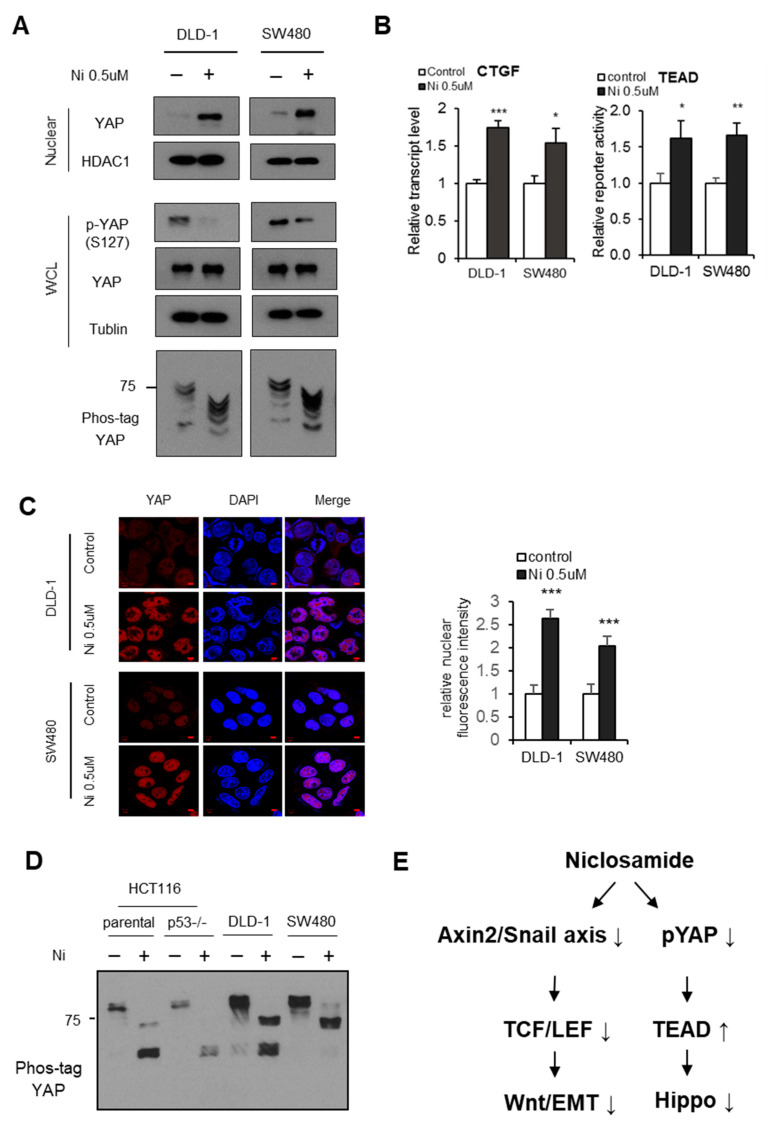
Niclosamide suppresses the canonical Wnt and Hippo pathways in APC-mutant cells. (**A**) Niclosamide inhibits the Hippo pathway through decreased YAP phosphorylation. Immunoblot analysis of endogenous YAP and phospho-YAP. YAP phosphorylation level in nuclear fraction and whole cell lysate (WCL) was determined by the pS127-YAP antibody and mobility shift on a phos-tag gel following niclosamide treatment (0.5 μM) for 16 h in CRC cells. HDAC1 and tubulin serve as loading controls for nuclear fraction and whole cell lysates (WCL), respectively. (**B**) Relative CTGF transcript abundance (left) and TEAD reporter activity (right) were determined by reporter assay and qRT-PCR, respectively. Statistical significances compared to control are denoted as * *p* < 0.05; ** *p* < 0.01; *** *p* < 0.001 by a two-tailed Student’s *t*-test. (**C**) The CRC cells were cultured in 80~90% density conditions and treated with niclosamide (0.5 μM); subcellular localization of endogenous YAP was determined by confocal microscopy. Relative nuclear YAP intensity was quantified using ImageJ and DAPI nuclear stain; scale bar, 5 μm. (**D**) HCT116 wild type (wt), HCT116 p53 null (p53 −/−), DLD-1, and SW480 cells were treated with 0.25 μM of niclosamide, and the phosphorylation of YAP was observed with a mobility shift phos-tag. The cells were grown in both serum- and glucose-starved condition and treated with niclosamide for 6 h before examination. (**E**) A schematic diagram of effects of niclosamide on the canonical Wnt and Hippo pathways by suppressing Axin2, Snail and phosphorylation of YAP.

**Figure 3 cancers-13-03437-f003:**
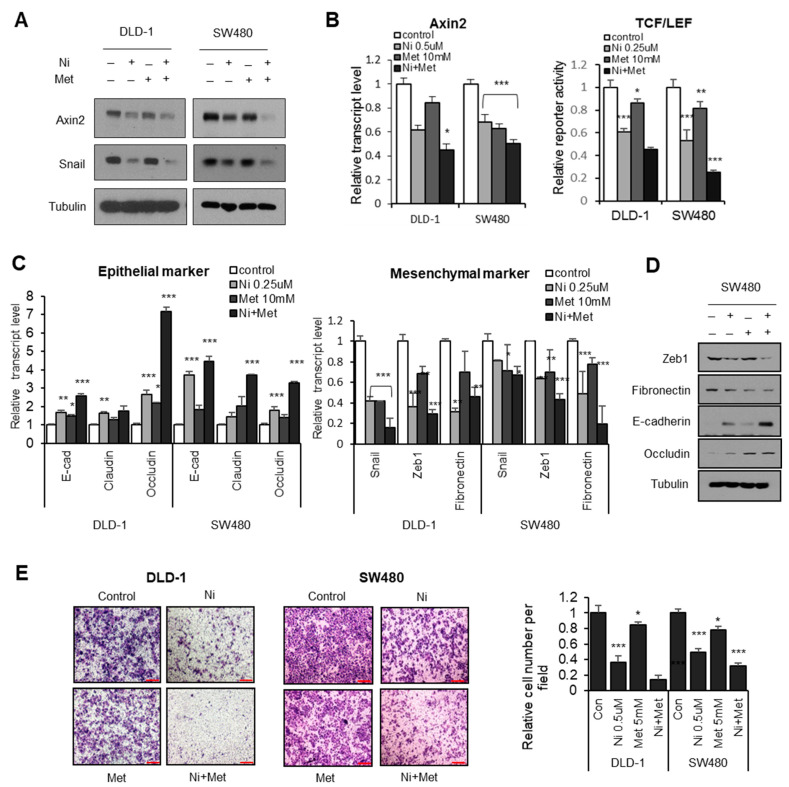
Metformin potentiates niclosamide in canonical Wnt and Snail-mediated EMT. (**A**) The CRC cells were treated with either niclosamide (0.5 μM), metformin (5 mM), or a combination of the two for 16 h; endogenous Axin2 and Snail protein abundance were determined using immunoblot analysis. (**B**) Relative Axin2 transcript abundance (left) and TCF/LEF reporter (TOP flash) activity (right) in CRC cells. Statistical significances compared to control are denoted as * *p* < 0.05; ** *p* < 0.01; *** *p* < 0.001 by a two-tailed Student’s *t*-test. (**C**) Relative transcript abundance of epithelial markers (left) and mesenchymal genes (right) were determined by qRT-PCR in CRC cells treated with either niclosamide (0.25 μM), metformin (10 mM), or a combination of the two for 16 h. Statistical significances compared to control are denoted as * *p* < 0.05; ** *p* < 0.01; *** *p* < 0.001 by a two-tailed Student’s *t*-test. (**D**) The protein levels of epithelial and mesenechymal markers were determined in SW480 cells. (**E**) The migration ability of colon cancer cells treated with niclosamide (0.5 μM), metformin (10 mM), or a combination of the two was determined by transwell migration assay; scale bar, 200 μm. Statistical significances compared to control are denoted as * *p* < 0.05; *** *p* < 0.001 by a two-tailed Student’s *t*-test.

**Figure 4 cancers-13-03437-f004:**
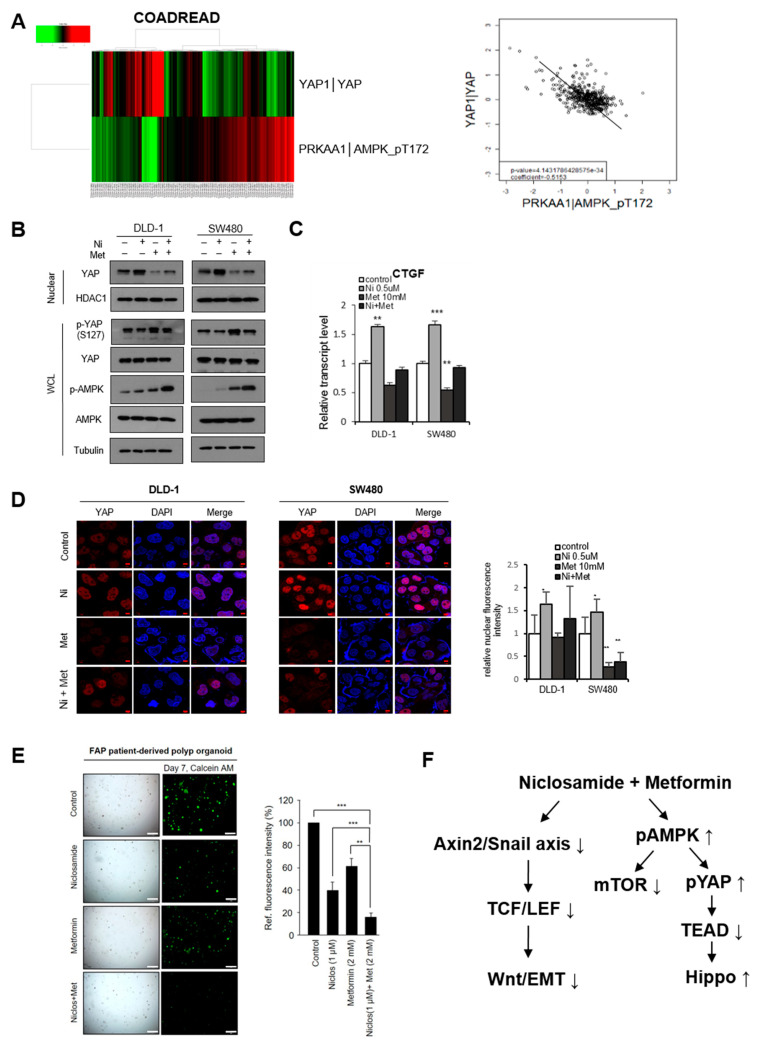
Metformin activates the Hippo pathway in CRC. (**A**) YAP abundance is inversely correlated to active AMPK in CRC samples. YAP and active AMPK (pT172) protein abundance in human COADREAD samples from the TCGA RPPA database is represented as a heatmap (left panel). Scatter plots of YAP and active AMPK (right panel, R2 = −0.5153, *p* = 4.143 × 10^−34^). (**B**) The colon cancer cells were treated with either niclosamide (0.25 μM), metformin (10 mM), or a combination of the two, and YAP and AMPK phosphorylation status in nuclear fraction and whole cell lysates (WCL) was determined by the pS127-YAP antibody. (**C**) Relative CTGF transcript abundance in CRC cells determined by qRT-PCR. (**D**) YAP abundance was determined by confocal immunofluorescence (left panels), and quantitative nuclear YAP intensity was determined using the ImageJ program (right panel); scale bar, 5 μm. Statistical significances compared to control are denoted as * *p* < 0.05; ** *p* < 0.01; *** *p* < 0.001 by a two-tailed Student’s *t*-test. (**E**) Patient-derived FAP organoids were established from biopsy tissue of FAP patient. The dissociated organoids were seeded in 20 μL of Matrigel and cultured with niclosamide (0.5 or 1 μM), metformin (2 mM), and their combination for 8 days. Organoids were then stained with calcein AM for live cells; scale bar, 500 μm. Quantitative analysis of live cell fluorescence intensities of organoids was performed using Image J. Means ± SD. * *p* < 0.05, ** *p* < 0.01, *** *p* < 0.001. (**F**) A schematic model for regulation of Wnt/EMT and Hippo with the combination of niclosamide and metformin in APC-mutated CRC.

**Figure 5 cancers-13-03437-f005:**
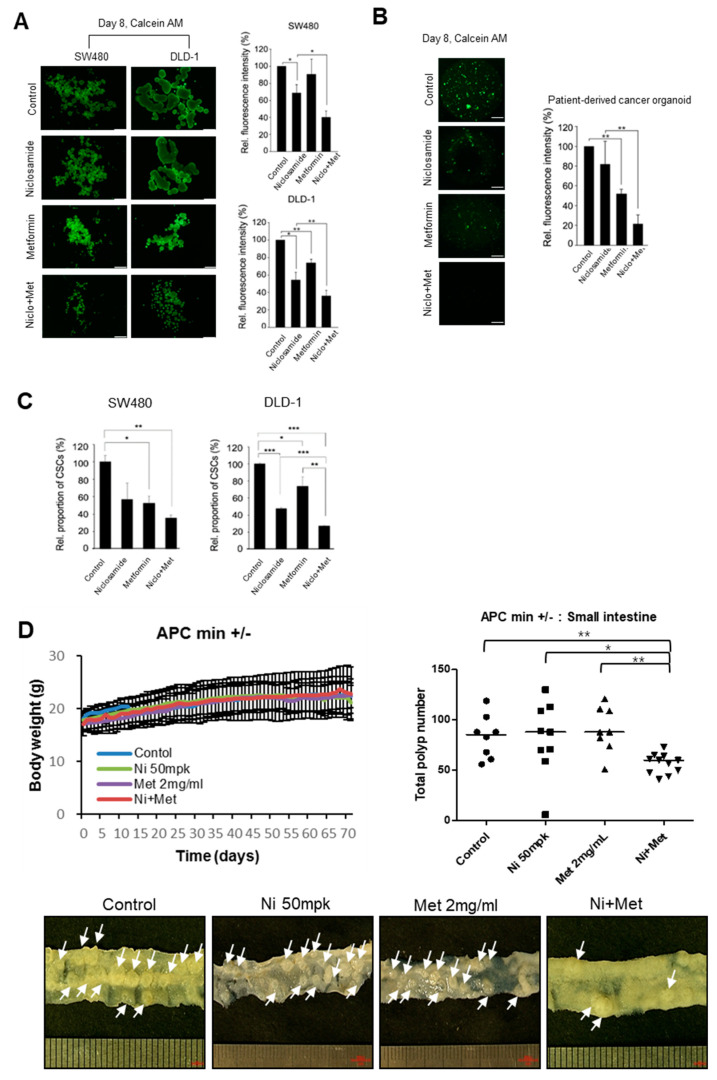
Combination of niclosamide and metformin suppresses tumor sphere, cancer stem cells, and organoids of CRC as well as the in vivo tumorigenic potential for colon cancer cells. (**A**) For tumor sphere culture, SW480 and DLD-1 cells were seeded in low attachment plates, cultured in sphere culture medium and treated with niclosamide (1 μM), metformin (2 mM), and a combination of the two for 8 days. Tumor spheres were then stained with calcein AM, and the live cell fluorescence intensities of tumor spheres were measured using ImageJ; scale bar, 500 μm. (**B**) Patient-derived cancer organoids were established from colon cancer biopsy tissue. The dissociated tumor organoids were seeded in 20 μL of Matrigel and cultured with niclosamide (0.5 μM), metformin (2 mM), and a combination of the two for 8 days. Organoids were then stained with calcein AM for live cells and quantitative analysis of the live cell fluorescence intensities of organoids was performed using Image J; scale bar, 500 μm. Means ± SD. * *p* < 0.05, ** *p* < 0.01, *** *p* < 0.001. (**C**) SW480 and DLD-1 cells were treated with metformin (10 mM), niclosamide (0.1 μM), and a combination of the two. After 48 h, CD44+/CD166+ cells were analyzed by flow cytometry using anti-FITC-CD44 and anti-PE-CD166. (**D**) APC-MIN mice were administrated with vehicle (control) or with either niclosamide (50 mg/kg, P.O.), metformin (2 mg/mL, P.O.), or a combination of the two daily until the 14-week endpoint. The body weight (upper left) and total number of polyps (upper right). Statistical significances compared to vehicle control was denoted as * *p* < 0.05; ** *p* < 0.01 by a Mann–Whitney test. Low power steromicroscopic images of representative intestine that showed the median adenoma value for each group. Arrows indicate adenoma foci in mouse intestine.

**Figure 6 cancers-13-03437-f006:**
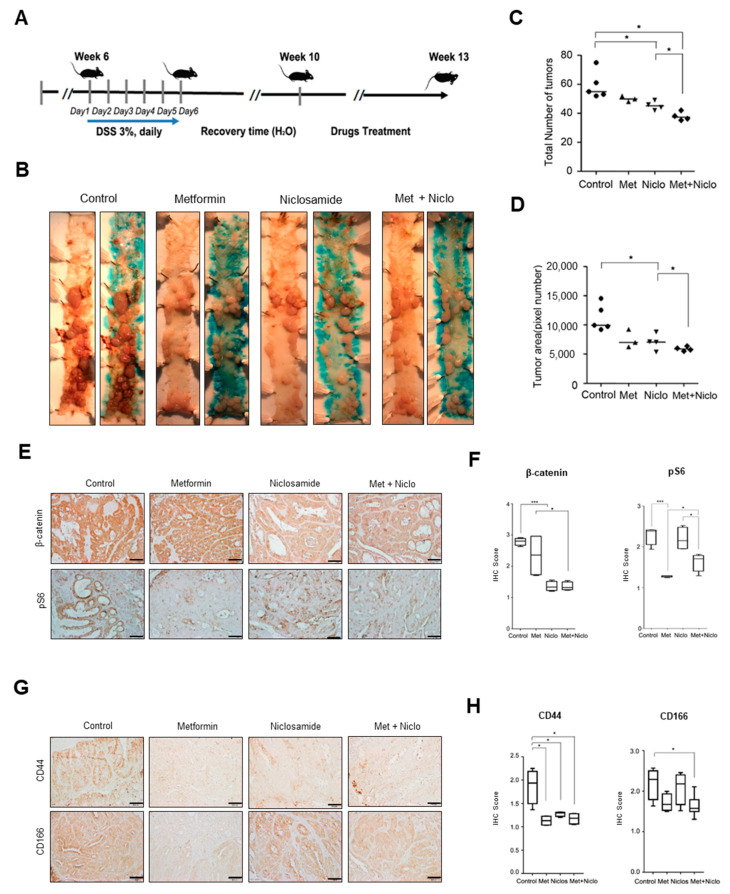
Combination of niclosamide and metformin effectively suppresses tumor progression in mouse colon cancer model. (**A**) Schematic diagram for DSS-induced CRC model using APC-MIN mice. (**B**) In mouse colon tumor model, the mice were evenly and randomly allocated into four groups; vehicle, metformin (metformin, 500 mg/kg/day in drinking water), niclosamide (niclosamide, 50 mg/kg/day in diet), and a combination of the same amounts of metformin and niclosamide; the mice were sacrificed after 3 weeks of drug treatment. (**C**,**D**) The number of tumors per mouse (**C**) and total tumor size (**D**) were compared, respectively, and quantitative analysis of total tumor size was performed using ImageJ. (**E**) Immunohistochemistry (IHC) of colon tumors treated with control, metformin alone, niclosamide alone, and metformin combined with niclosamide was performed on sections of formalin-fixed, paraffin-embedded, dissected tumor sample to determine β-catenin and pS6; scale bar, 200 μm. (**F**) In the stained colon tumor, the expression of β-catenin and pS6 was evaluated using IHC scores based on the ImageJ program. (**G**,**H**) IHC was performed for CSC markers (CD44 and CD166) and analyzed in the same manner as above in (**E**,**F**); scale bar, 200 μm. Data are expressed as the means ± SD. n = 5 per group. * *p* < 0.05, *** *p* < 0.001.

## Data Availability

The data presented in this study are available in this article and in the Appendix A.

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
