# Peer review of "Metformin and Niclosamide Synergistically Suppress Wnt and YAP in APC-Mutated Colorectal Cancer"

_cancers, 2021, doi:10.3390/cancers13143437_

Round 1
Reviewer 1 Report
The manuscript entitled “Metformin and niclosamide synergistically suppress Wnt and YAP in APC-mutated colorectal cancer” underlies the anticancer potential of the niclosamide and metformin combinatorial approach through the suppression of Wnt and YAP activity in APC-mutated colorectal cancer cells. The study was properly performed and the results are clearly presented. However, the following concerns have to be addressed before the paper can be accepted for publication.
Major revisions:
- The authors showed that “Axin2 transcript abundance in CRC patient samples was inversely correlated to YAP target genes CTGF and CRY61” and they performed these analyses in the APC or CTNNB1 mutant samples. It would be interesting to interrogate other datasets (i.e. Gepia) in order to analyze the expression levels of Axin2, CTGC and CRY61 in CRC samples respect to normal colon tissues. Moreover, are these markers predictive of poor prognosis in CRC patients?
- It would be interesting to evaluate the protein levels (by western blotting) of Axin2, YAP and p-YAP in cells not APC-mutant and, if possible, also in control (not-tumor) cells.
- From the immunofluorescences in figures 2C and 4D the nuclear localization of YAP is unclear. It would be better to merge the images (DAPI and YAP staining) to better analyze the translocation.
- To confirm that Niclosamide and Metformin treatments modulate EMT, the authors must analyze the expression of the epithelial and mesenchymal markers also at protein levels (i.e. western blotting).
- The authors declare that Ni and Met treatments reduced the CD44/CD133 expression, and therefore target the cancer stem cells (CSCs) populations. The author must show the FACS plot in order to evaluate the reduction of CD44 and CD133 induced by the treatment.
- Recently, Delle Cave et al., demonstrated that the L1CAM positive/CXCR4 positive double population shows enhanced stemness, migratory and metastatic capacity. Please check the expression levels of L1CAM and CXCR4 in the cells after treatment with Ni and Met and cite this important paper (doi:10.7150/thno.54027). I will aspect a reduction in L1CAM and CXCR4 expression.
- In the paragraph 3.5 the authors used another CSC marker for the IHC, CD166, instead of CD133. Why do the authors choose this marker? The immunohistochemistry of CD133 needs to be added.
Minor revisions:
- The quality of all images is poor. Please change it. Furthermore, the size bar is missing in both IF and IHC figures. Please add it.
- A thorough rereading of the text is necessary, as some sentences are not correct in English. Moreover, some section (i.e. 2.1, 3.1 and Discussion) begin with a general description that needs to be removed. In addition, in the 2.11 part the “b” of “-catenin” has been lost. Finally, there is a mistake in the 2.2. part. The authors declare that “Niclosamide and metformin were treated for 16 hours, then washed twice with PBS, etc..” but they obviously refer to cells, not molecules. This statement must be corrected.
Reviewer 2 Report
The authors report that the Wnt signaling target Axin2 is an activator of YAP phosphorylation in APC-mutated colorectal cancer (CRC). They used a combination of clinically available niclosamide and metformin to suppress Wnt signaling and simultaneously activate Hippo signaling for a better treatment of CRC in cultured cells, organoid, and an APC-MIN mouse model. The main results seem technically solid and interesting. Several concerns should be addressed.
- The changes of several key Wnt signaling mediators that are interactive or closely related to Axin2, including Gsk3b, beta-catenin, and downstream Tcf/Lef1 activities should be addressed in various drug treatments in this study.
- Figures 2E and 4F are problematic. inhibited Axin2/Snail unlikely diminishes Tcf/Lef1 activities, which is also not demonstrated in this study. It also reads that repressed pYAP upregulates TEAD and inhibits Hippo in 2E, but it downregulates TEAD and activates Hippo in 4F, which are very confusing or misleading.
- 4E should also show brightfield pictures of organoids.
- 5C shows no significance of Nicol+Met treatments?
- 6H should label Y axis.
- Method 2.11 shows only “-catenin” in the text.
Round 2
Reviewer 1 Report
I think the authors greatly addressed all points of the revision.
However, it seems to me that only one point is missing:
I can not see the scale bar in the immunofluorescence images in FIgure 2 C and Fgure 4 D. Please check that.
Author Response
We appreciate the helpful suggestions. We replaced the scale bars in the immunofluorescence images in Figure 2C and Figure 4D with larger ones.
Reviewer 2 Report
Most of the previous concerns have been properly addressed.
Author Response
We appreciate the helpful review.